# Recapitulating the Key Advances in the Diagnosis and Prognosis of High-Grade Gliomas: Second Half of 2021 Update

**DOI:** 10.3390/ijms24076375

**Published:** 2023-03-28

**Authors:** Guido Frosina

**Affiliations:** Mutagenesis & Cancer Prevention Unit, IRCCS Ospedale Policlinico San Martino, Largo Rosanna Benzi 10, 16132 Genova, Italy; guido.frosina@hsanmartino.it; Tel.: +39-010-555-8543

**Keywords:** CNS, diagnosis, glioma, MRI, neuroradiology, non-imaging diagnosis, nuclear medicine, prognosis, tumor

## Abstract

High-grade gliomas (World Health Organization grades III and IV) are the most frequent and fatal brain tumors, with median overall survivals of 24–72 and 14–16 months, respectively. We reviewed the progress in the diagnosis and prognosis of high-grade gliomas published in the second half of 2021. A literature search was performed in PubMed using the general terms “radio* and gliom*” and a time limit from 1 July 2021 to 31 December 2021. Important advances were provided in both imaging and non-imaging diagnoses of these hard-to-treat cancers. Our prognostic capacity also increased during the second half of 2021. This review article demonstrates slow, but steady improvements, both scientifically and technically, which express an increased chance that patients with high-grade gliomas may be correctly diagnosed without invasive procedures. The prognosis of those patients strictly depends on the final results of that complex diagnostic process, with widely varying survival rates.

## 1. Introduction

High-grade gliomas (HGG; World Health Organization grades III and IV) are invariably fatal brain tumors that are classified as “rare” cancers, with an incidence of fewer than 15 out of 100,000 people each year [1,2]. Nevertheless, their global burden in absolute terms is at least 15,000 casualties every year in Europe only [3]. Since 2005, when temozolomide was introduced into their therapy [4], HGG patients have suffered from a lack of effective novel therapeutic options, with the median survival rates being disappointingly steady at 24–72 and 14–16 months for grades III and IV, respectively [5]. On the contrary, the diagnosis and prognosis of these tumors has gradually improved both technically and scientifically in the last decade [6]. We have examined the most recent advances in the diagnosis and prognosis of HGG that were published during the second half of 2021. We confirm that the research activity in the field is fruitful, with novel technological and scientific options that promise to accelerate and improve the resolution of the diagnosis and prognosis of HGG patients.

## 2. Methods

A literature search in PubMed was performed using the general terms “radio* and gliom*” and a time limit from 1 July 2021 to 31 December 2021. Of the 742 automatically identified records, 532 were manually removed because they did not meet the relevance criteria using the terms “Radiology of HGG” or “Radiotherapy of HGG” based on the titles of the articles. Of the remaining 210 records, another 54 were also deemed to be unsuitable based on the content of the article abstracts. The pdf files of the resulting 156 articles were all retrieved. Based on their content, 20 of them were discarded as they out of our scope, while the remaining 136 were further analyzed and manually assigned to the “Radiology of HGG” group (*n* = 62, discussed here) or “Radiotherapy of HGG” group (*n* = 74, to be discussed in another article). In addition to the 62 articles published in 2021, for the purpose of clarity and completeness, this review includes 28 additional relevant articles published outside of the year 2021 with a total of 90 references. A relevant flowchart is shown in Figure 1. 

## 3. Advances in Diagnosis

### 3.1. The 2021 WHO Classification of Tumors of the CNS

Traditionally, brain tumors have been classified based on their morphological features, with a significant variability in patient survival even within the same histological grade. Over recent years, advances in molecular profiling have changed the diagnostic approach and classification of brain tumors, leading to the development of an integrated morphological and molecular classification endowed with improved clinical relevance [8] (Table 1). Our expanding knowledge of brain tumor genetics and the development of new technologies required a number of updates in diagnosis and prognosis, which were regularly published by the Consortium to Inform Molecular Practical Approaches to CNS Tumor Taxonomy–Not Official WHO (c-IMPACT-NOW). The fifth edition of the WHO classification of CNS tumors in 2021 systematically organized the literature [9]. Building on the fourth edition from 2016, it introduced major advances on the role of molecular diagnostics in CNS tumor classification. At the same time, it remained wedded to other established approaches to tumor diagnosis, such as histology and immunohistochemistry (IHC), providing different approaches to both CNS tumor nomenclature and grading and emphasizing the importance of integrated diagnoses. New tumor types and subtypes were introduced, and some of them are based on technologic advances such as next-generation sequencing (NGS) and DNA methylome profiling, with over 40 tumor types and subtypes eventually being defined by their key molecular features [9]. To merely note a couple of examples, the accurate detection of genetic variants such as single substitutions (in *idh1/2* and *tert* genes), chromosomal abnormalities (*CDKN2A*, *1p/19q* deletions, and *EGFR* amplifications), and promoter methylations (*MGMT*) became critical for HGG patient management, and new techniques such as quantitative polymerase chain reaction (qPCR) and NGS became more reliable with respect to the reliability of older ones such as IHC [10]. This had consequences on treatment planning as well; based on molecular profiling, the American Society for Radiation Oncology (ASTRO) task force proposed recommendations concerning the RT management of molecularly characterized HGG patients, including the possible use of intensity-modulated RT/volumetric-modulated arc therapy or proton therapy [11]. Concerning CNS tumors other than HGG, aggregated molecular evidence points towards the fusion of *zfta* or *yap1* genes in the development of supratentorial ependymomas; *h3*, *ezhip*, or *tert* mutations in some infratentorial ependymal tumors; *mycn* amplifications in spinal ependymomas, in addition to the previously known mutations in *nf2* [12,13].

HGGs are major contributors to oncologic morbidity and mortality in the adolescent and young adult (AYA) demographics as well [14]. The favorable impact of a younger age and a higher Karnofsky Performance Status (KPS) on overall survival (OS) is established for AYA patients. A positive association of OS with *idh* mutant molecular status and the extent of surgical resection is also demonstrated, though many studies may improperly include adults as well. Hence, albeit a number of genetic profiles have been characterized, the standard for their application to AYA populations is less clear than it is in adults, and there is a major need to develop more robust evidence-based practices for the diagnosis and management of AYA HGG patients [14].

**Table 1 ijms-24-06375-t001:** Highlights of recent advances in the diagnosis of HGG.

Major Finding	Experimental System	LE ^1^	Ref
*The 2021 WHO Classification of CNS Tumors*			
The fifth edition of the world health organization (WHO) classification of central nervous system (CNS) tumors in 2021 introduced advancements in the role of molecular diagnostics in CNS tumor classification. At the same time, it remained wedded to other established approaches to tumor diagnosis, such as histology and immunohistochemistry (IHC), paving the way for integrated approaches to both CNS tumor nomenclature and grading.	New CNS tumor types and subtypes were classified, and some were based on technological advances such as next-generation sequencing (NGS) and DNA methylome profiling. Over 40 tumor types and subtypes were eventually defined.	IV	[9]
*Advances in imaging diagnosis of HGG*			
*Advances in MRI*			
The relative cerebral blood volume (rCBV) and percentage signal recovery (PSR) ratio are different in HGG and primary central nervous system lymphoma (PCNSL) at both the group level and subject level. Incorporation of perfusion in routine magnetic resonance imaging (MRI) of contrast-enhancing lesions may have an impact on patient management.	Patients who underwent diagnostic imaging for a primary brain lesion were retrospectively examined. Fifteen immunocompetent patients with PCNSL containing a total of 15 lesions, and 11 patients with high-grade glioma (HGG), containing a total of 11 lesions, were identified and analyzed.	VIII	[15]
Detailed MRI data on *h3k27m* mutant diffuse midline glioma (DMG) pediatric patients were provided.	A total of 85 pediatric patients with DMG were retrieved from registries. Histopathological and molecular genetic data, including *h3k27* mutation status, as well as MRI studies of diagnosis, were compared.	VIII	[16]
Secondary gliosarcoma (SGS) components likely have a monoclonal origin, and the clone possessing mutations in *NF1* and *TP53* was likely the founding clone in a case of SGS.	Case report: Somatic mutation profiles in a glioblastoma (GB), and then an SGS patient, were examined using whole-genome sequencing and deep-whole-exome sequencing. Mutation signatures were characterized to investigate the relationship between chemo-RT and SGS pathogenesis.	VIII	[17]
Combined analysis of susceptibility-weighted imaging (SWI) and diffusion-weighted imaging (DWI) could differentiate atypical GB from PCNSL and distinguish major genomic subtypes between these tumors.	Thirty-one immuno-competent patients with PCNSL stratified by *BCL2* and *MYC* rearrangement, and fifty-seven patients with atypical GB (no visible necrosis) grouped according to *idh1* mutation status underwent 3.0-T MRI before treatment in this retrospective study.	VIII	[18]
A transfer learning approach suggested that convolutional neural networks may provide accurate diagnostic information to assist radiologists in distinguishing PCNSL from GB.	Preoperative brain tumor MRIs were retrospectively analyzed among 320 patients with either GB or PCNSL from two academic institutions.	VIII	[19]
A machine learning classifier based on the analysis of longitudinal perfusion time points and combined structural and perfusion features may enhance the classification outcome.	Study participants were separated into two groups: group I had a single dynamic susceptibility contrast (DSC) time point, and group II had longitudinal DSC time points. Structural MRI and DSC-MRI scans were retrospectively analyzed.	VIII	[20]
DSC–MRI radio genomics in HGG may allow the increased predictive performance of *idh* mutation status.	A multicenter study featuring an exploratory set for radiomics model development and external validation using two independent cohorts.	VIII	[21]
AC/DC coils might be used synergistically with optimized acquisition schemes to improve metabolic imaging in HGG patients. This methodology may be applicable to other neurological disorders.	Four HGG patients and five volunteers that underwent 3.0 T MRI were analyzed.	VIII	[22]
Diffusion variance decomposition (DIVIDE) imaging is a promising technique for glioma characterization and diagnosis.	Ninety-three participants with suspected glioma according to preliminary CT and MRI results were analyzed.	VIII	[23]
Training a classifier to predict both *idh*-mutation and *1p/19q*-codeletion status outperformed a tiered structure that first predicted *idh*-mutation, then *1p/19q*-codeletion. Including apparent diffusion coefficient (ADC), a surrogate marker of cellularity, more accurately detected differences between subgroups.	Three hundred and eighty-four patients with newly diagnosed gliomas who underwent preoperative MRI with standard anatomical and DWI and one hundred and forty-seven patients from an external cohort with anatomical imaging were analyzed.	VIII	[24]
Amide proton transfer (APT) imaging can be used to differentiate HGG from low grade gliomas (LGG) in pediatric patients and provide added value beyond quantitative relaxation times.	In this prospective study, APT imaging and relaxation time mapping were performed in 203 pediatric patients suspected of having gliomas.	VII	[25]
Brainstem glioma (BSG) patients with *h3k27m* mutant had higher max APT values than the wild-type patients did. APT-derived radiomics could accurately predict a *h3k27m* mutant status in BSG patients.	Eighty-one BSG patients with APT imaging at 3.0-T MRI and known *H3K27M* status were retrospectively studied.	VIII	[26]
*h3.1k27m*-mutated tumors have higher ADC and lower perfusion values than *h3.3k27m* ones do, without direct correlation with microvascular or nuclear density.	Twenty-seven treatment-naïve children with histopathologically confirmed *h3k27m* mutant diffuse intrinsic pontine glioma (DIPG) were prospectively analyzed.	VIII	[27]
Dynamic nuclear polarization (DNP)-MRI might represent a useful technique with which to evaluate HGG metabolism before and after RT in the clinical setting.	DNP-MRI chemical shift imaging with hyperpolarized [^1−13^C] pyruvate was conducted to evaluate the metabolic change in glycolytic profiles after radiation of two glioma initiating cell (GIC)-driven (GBMJ1 and NSC11) and an adherent human HGG cell line (U251) in an orthotopic xenograft mouse model.	VIII	[28]
Sequential administration of 5-aminolevulinic acid (ALA) and iron supplements increases the iron deposition in HGG cells, enabling clinical 3.0-T MRI to detect HGG using R2′ or quantitative susceptibility mapping (qSM).	Intra-cellular iron accumulation in HGG cells treated with ALA and/or ferric ammonium citrate (FAC) was measured. Cell phantoms containing HGG cells and Wistar rats bearing the C6 HGG were imaged using a 3.0-T MRI scanner after sequential administration of ALA and FAC. Relaxivity and qSM analysis were performed on the images.	VIII	[29]
*Advances in PET*			
Although L1 amino acid transporter (LAT1) is reported to mediate the uptake of O-(2-^18^F-fluoroethyl)-L-tyrosine (^18^F-FET) into tumour cells, the levels of LAT1 expression do not correlate with the levels of ^18^F-FET uptake in *idh* mutant HGG. In particular, the lack of tracer uptake in ^18^F-FET-negative HGG cannot be explained by a reduced LAT1 expression.	Forty newly diagnosed *idh* mutant HGG without *1p/19q* codeletion were evaluated: (*n* = 20 ^18^F-FET-negative (tumor-to-background ratio (TBR) < 1.6) and *n* = 20 ^18^F-FET-positive (TBR > 1.6) HGG.	VIII	[30]
Residual 3,4-dihydroxy-6-^18^F-fluoro-L-phenylalanine (^18^FDOPA) hypermetabolic burden predicted overall survival (OS) for isocitrate dehydrogenase *(IDH)* wild-type gliomas, regardless of the tumor grade.	Thirty-four patients with treatment-naïve *IDH* wild-type gliomas (WHO grade II 6, III 15, and IV 13) were retrospectively included.	VIII	[31]
Radiomics based on time-to-peak (TTP) images extracted from dynamic ^18^F-FET PET can predict the *ptert*-mutation status of *IDH* wild-type diffuse astrocytic HGG preoperatively.	A total of 159 patients with newly diagnosed *IDH* wild-type diffuse astrocytic HGG and dynamic ^18^F-FET PET prior to surgical intervention were enrolled and randomly divided into training (*n* = 112) and testing (*n* = 47) cohorts.	VII	[32]
Tumor isocontour T-maps and the combined analysis of cerebral blood flow (CBF) and ^18^F DOPA-PET uptake resulted in diagnosis for differentiating between progression and pseudo-progression in treated gliomas. The sensitivity is particularly high for GBs.	Fifty-eight patients with treated unilateral gliomas who were included in the study constituted groups with ten (17.2%) LGG and forty-eight (82.8%) HGG patients.	VIII	[33]
In lymphomatosis and gliomatosis, fluorodeoxyglucose (FDG) accumulates in only part of the lesion. FDG is thus less suitable than MET is for depicting those lesions.	Ten patients had lymphomatosis or gliomatosis of the brain. The underlying pathologies included intravascular lymphoma (*n* = 1), lymphomatosis (*n* = 3), and gliomatosis (*n* = 6).	VIII	[34]
A possible step forward in the clinical translation of cancer vaccination as a potential HGG therapy, as well as having the benefits of monitoring efficacy of these treatments using immunoPET imaging of T cell activation.	A subcutaneous vaccination approach with CpG oligodeoxynucleotide, OX40 mAb, and tumor lysate at a remote site in a murine orthotopic HGG model was developed to induce the activation of T cells distantly, while monitoring their distribution in stimulated lymphoid organs with respect to the observed therapeutic effects.	VIII	[35]
*Advances in non-imaging diagnosis of HGG*			
Significant differential expression of genes involved in cancer inflammation and immunity crosstalk were found among patients with different glioma grades, and there was positive correlation between their transcriptomic profiles in the plasma and tumor samples and with the cancer genome atlas (TCGA) glioma-derived RNA.	Blood samples were collected from twenty glioma patients prior to tumor resection. Plasma circulating cell-free messenger ribonucleic acids (ccfm-RNAs) and glioma-derived RNA were extracted and profiled.	VIII	[36]
Brain activity was significantly increased in the tumor hemisphere in general and in peritumoral regions specifically. However, none of the measures and spatial levels of brain activity correlated with changes in tumor volume, nor did they differ between patients with increasing versus stable tumor volumes.	The relationship between brain activity and increasing tumor volumes on routine clinical MRI in glioma patients was assessed. Postoperative magnetoencephalography (MEG) was recorded in 45 diffuse glioma patients.	VIII	[37]
The combination of a HGG, multiple development venous anomalies (DVAs) and malformations of cortical development (MCD) in a paediatric or young adult patient should prompt the neuroradiologist to hypothesize an underlying diagnosis of constitutional mismatch repair deficiency (CMMRD).	Case report: an 8-year-old boy with acute headache, vomiting, and an episode of unconsciousness in whom brain imaging revealed an HGG.	VIII	[38]
The most common mismatch repair deficiency (MMRD) primary brain tumor was GB *IDH* wild-type. The genetic profile of MMRD GB was different from that of conventional GB.	Thirteen MMRD-associated (nine sporadic and four Lynch syndrome) primary HGGs were analyzed to determine their clinicopathological, molecular characteristics, and biological behavior.	VIII	[39]
A pathological diagnosis can be made safely and efficiently in brainstem lesions using stereotactic biopsy.	The medical records of 42 adult patients who underwent stereotactic biopsy on brainstem lesions were retrospectively analyzed.	VIII	[40]
Radiation-induced organizing hematoma (RIOH) is more likely to occur earlier with thick tumor wall in subjects who underwent gamma knife surgery (GKS) than it is in patients who underwent conventional radiotherapy (RT). These results indicate the clinical course of RIOH differs based on type of treatment and might help determine the duration of follow-up.	Thirty-seven RIOHs confirmed by surgical excision were divided into subgroups based on type of radiation treatment and pathology of the original lesion. The clinicopathological results were compared between the groups.	VIII	[41]

^1^ Level of evidence according to https://guides.library.stonybrook.edu/evidence-based-medicine/levels_of_evidence (accessed on 14 March 2023).

### 3.2. Advances in MRI

The preoperative differentiation of PCNSL from HGG is important, as surgical and adjuvant therapies may be fundamentally different between the two types of tumors. Several imaging techniques have been explored to distinguish the two neoplasms preoperatively. The ability of the dynamic susceptibility contrast (DSC)-derived metrics, percentage signal recovery (PSR), and relative cerebral blood volume (rCBV) to distinguish between PCNSL and HGG has been assessed by Chaganti and co-workers [15]. In all, 15 immuno-competent PCNSL patients and 11 HGG patients were retrospectively analyzed. The histological characteristics were correlated with the perfusion data (in particular, the rCBV and PSR parameters) to determine the sensitivity, specificity, and diagnostic capacity of the perfusion techniques in comparison to those of classical histology (Figure 2a–m). Significant differences in the rCBV and PSR values were observed between the two groups of PCNSL and HGG patients. Notably, the mean values of rCBV and PSR were lower and higher in PCNSL than they were in HGG, respectively. For rCBV, a value less than 2.68 was predictive of a PCNSL diagnosis rather than HGG, with the amazing sensitivity and specificity of 100%. For the PSR, a value greater than 0.9 was also predictive of a PCNSL diagnosis rather than HGG, with sensitivity and specificity rates of 100% and 90.91%, respectively. Hence, the rCBV and PSR ratios are different in PCNSL and HGG at both the group level and subject level, and the incorporation of perfusion in MRI examinations of contrast-enhancing lesions may contribute to the patient’s management [15]. Similarly, concerning the differentiation of PCNSL and HGG, Ozturk and co-workers [18] have reported that susceptibility-weighted imaging (SWI) along with diffusion-weighted imaging (DWI) may allow the differentiation of PCNSL from HGG and even distinguish major genomic subtypes between these tumors [18]. Finally, a convolutional neural network on contrast-enhanced T1-weighted images has been developed by Mc Avoy and co-workers [19]. Despite the limited numerical size, this study suggested that a convolutional neural network may significantly contribute to the neuroradiologist’s differential diagnosis of PCNSL versus GB.

Siakallis and collaborators [20] developed an automatic classifier that uses a combination of MRI and perfusion images to distinguish between disease stability, progression, and pseudo progression. The results showed that this classifier might improve differential diagnoses [20].

T2-fluid attenuated inversion recovery (FLAIR) MRI-based radiomics permits the identification of isocitrate dehydrogenase (*idh*) mutant and *1p/19q* co-deleted HGG [42], while DSC MRI-based radiomics does not seem to be capable of predicting *idh* mutations in HGG. The technical reasons underlying this difference have been discussed by Manikis and co-workers [21]. Albeit T2-FLAIR has been suggested to be 100% specific for *idh* mutant *1p/19q* non-codeleted HGG, false positives have been recently reported [43]. The criteria and potential pitfalls for the application of T2-FLAIR have been discussed by Pinto and co-workers [44]. In particular, the imaging of D-2-hydroxyglutarate produced by HGG *idh* mutations may be limited by lipid artifacts in tissues surrounding the brain [22]. Some technical tricks to possibly improve the resolution of D-2-hydroxyglutarate maps have been proposed by Strasser and co-workers [22]. The diffusion variance decomposition (DIVIDE) technique has been used by Li and co-workers for the molecular and microstructural classification of HGG [23]. It was proposed that DIVIDE may effectively distinguish low-grade gliomas (LGG) from HGG and *idh*-mutated from wild-type tumors. According to Cluceru and colleagues [24], the simultaneous radiomics analysis of both the *idh* mutation and the *1p/19q* codeletion may be more effective than the subsequent level analysis is, i.e., performed first for the *idh* mutation, and then for the *1p/19q* codeletion.

Whether amide proton transfer (APT) MRI can be used to characterize HGG in pediatric patients was evaluated by Zhang and co-workers [25]. In particular, APT MRI showed significantly higher values in pediatric patients with HGG than it did those with LGG, which could be useful to differentiate tumor gradings [25]. APT imaging and radiomic features were further used by Zhuo and co-workers to predict the tumor *h3k27m* mutation status [26]. Patients with tumors with the *h3k27m* mutation had higher maximum APT values than those with wild-type tumors did, suggesting that APT-based radiomics might help in determining the presence of an *h3k27m* mutation [26]. The pediatric DIPGs may now be better characterized on the basis of the histone *h3* mutational status. The latter type was correlated with the tumor histological and molecular characteristics by Calmon and co-workers [27].

Better overall survival (OS) and younger age were found in *h3.1k27m* versus *h3.3k27m* mutations-bearing DIPG patients. Further, *h3.1k27m*-mutated tumors showed higher apparent diffusion coefficient and lower perfusion values than the *h3.3k27m*-mutated tumors did, possibly due to tissue edema variations [27]. Detailed MRI data on *h3k27m* mutant pediatric diffuse midline gliomas with respect to the molecular subgroup status and site of development have been provided by Hohm and co-workers [16]. Most of the *h3k27m* mutants were found in the pons and thalamus/basal ganglia (Figure 2n–s), while the *H3K27* wild-type tumors had more heterogeneous localizations. Diffuse hemispheric gliomas with histone *h3g34r* mutations have recently been characterized both clinically and neuroradiologically [45]. These mutant tumors are found more frequently in supratentorial brain regions of adolescent patients. The prognosis is variable and partly dependent on the status of the tumor margins.

Rykkje and colleagues discussed the best timing for performing post-surgical MRI to evaluate the extent and effectiveness of surgical resection [46]. Increases in contrasts unrelated to the neoplastic tissues were observed at all times following surgery and even during surgery (intraoperative MRI). The least artificial contrast was observed 1–2 days after surgery, which is considered to be the most appropriate time to perform the post-surgical control via MRI [46].

Dynamic nuclear polarization (DNP)-MRI with hyperpolarized ^1-13^C pyruvate has been used to evaluate the metabolic change in glycolytic profiles after radiotherapy (RT) of multiple orthotopic HGG animal models [28]. DNP-MRI permitted the evaluation of lactate dehydrogenase-A expression before and after radiation [28].

Ebrahimpour and colleagues have perfected the method of the visualization of tumor tissue based on the administration of 5-aminolevulinic acid (ALA) via an iron supplement [29]. The administration of the iron supplement after that of ALA allowed the iron deposition to increase in the tumor cells, increasing the resolution of the 3.0 T MRI.

After GB treatments, the onset of secondary sarcomatous components (secondary gliosarcoma—SGS) occurs, albeit rarely. The origin of these components is unclear. One possibility is that they are induced by RT, which can actually induce sarcomas in patients who have other types of cancer. In one of those rare cases of the gliosarcomatous development of a GB, Li et al. found that the sarcoma likely resulted from a single mutant clone in the *nf1* and *tp53* genes [17]. A preoperative MRI showed an edematous lesion in the right frontal and parietal lobes (Figure 2t). The lesion was surgically removed, and after histological analysis, it was found to be a GB with mutations in the *pten* and *tp53* genes and *egfr* amplification (Figure 2u). Three weeks later, RT as per the guidelines began and continued for a duration of 6 weeks, followed by chemotherapy with temozolomide (TMZ). A brain MRI performed 3 months after the initial resection showed a new lesion located in the same site of the original tumor (Figure 2t). The patient was treated in second line with bevacizumab plus dose-dense TMZ, but the symptoms rapidly worsened, and a second surgery had to be carried out. The histopathological diagnosis following the second surgery identified an SGS-expressing GFAP and reticulin-rich sarcomatous elements (Figure 2v). Other molecular and genetic characteristics of the SGS tissue were similar to those of the primary GB.

### 3.3. Advances in PET

O-(2-^18^F-fluoroethyl)-L-tyrosine (^18^F-FET) is a widely used PET tracer in HGG metabolic imaging. It has been suggested that ^18^F-FET uptake is linked to the action of L1 amino acid transporter (LAT1). However, 5% of HGG do not capture ^18^F-FET (“^18^F-FET-negative gliomas”), and the pathophysiological basis of this phenomenon has been studied by Vettermann et al. [30]. No relationship was observed between the LAT1 levels and tumor uptake capacity, suggesting that the possible lack of ^18^F-FET uptake is not due to the poor expression of LAT1. The association between the biological tumor volume, as determined by different techniques including contrast-enhanced volume on T1-weighted images, FLAIR hyperintense volume, and 3,4-dihydroxy-6-^18^F-fluoro-L-phenylalanine (^18^FDOPA) hypermetabolic volume and the OS of *IDH* wild-type HGG patients has been studied by Tatekawa and co-workers [31]. Regardless of the tumor grade, ^18^FDOPA hypermetabolic volume best predicted the OS of patients with *IDH* wild-type gliomas [31].

The use of static and dynamic PET imaging using ^18^F-FET as radio metabolite has been proposed by Li et al., for the identification of *ptert* mutations in patients with wild-type *IDH* HGG. The dynamic PET investigation was suggested to permit the determination of the prognostically informative *ptert* mutation status in those patients [32].

The use of PET-coupled perfusion with ^18^FDOPA was used by Pellerin and collaborators for the follow-up of HGG patients [33]. The combined analysis of cerebral blood flow and absorption and metabolism by ^18^F-DOPA PET allowed them to discriminate between progression and pseudo progression with greater precision than the precision when the two techniques were used separately.

The use of the radio metabolites ^11^C-methionine and ^18^F-fluorodeoxyglucose for the differential diagnosis of PCNSL or HGG by PET has been studied by Tomura and collaborators [34] (see also the above “MRI” section). It turned out that ^18^F-FDG is less suitable for this type of investigation than ^11^C-methionine is, since the former, unlike the latter, accumulates only in a part of the lesion.

The potential clinical application of unconventional non-amino acid PET radio-pharmaceuticals in patients with HGG was discussed by Laudicella and co-workers [47]. Novel investigational PET radiopharmaceuticals deserve to be further explored in studies specifically designed to validate the preliminary findings [47]. In particular, OX40 is a molecule expressed in effector T cells and is used as a biomarker to evaluate the immunological response of the HGG patient by immune PET with ^89^Zr-DFO-OX40 monoclonal antibodies. This type of immune PET imaging of T cell activation may be useful to monitor the efficacy of immunotherapies for HGG [35].

### 3.4. Advances in Non-Imaging Diagnosis

Ita and collaborators studied whether there are significant differences in the RNAs present in the plasma of HGG patients in comparison to those in healthy subjects [36]. The analysis of the transcription profiles in plasma samples revealed that several genes were significantly over expressed in HGG patients as compared to the levels in healthy controls, including *BCL2L1*, *GZMB*, *HLA-A*, *IRF1*, *MYD88*, *TLR2,* and *TP53* (Figure 3a). The *BCL2*, *CCR2*, *CXCL9*, *CXCR3*, *GBP1*, *HIF1A,* and *IL23A* genes were, on the contrary, significantly under-expressed in HGG patients compared to the levels in the control group (Figure 3b). The significant differential expressions of genes involved in cancer inflammation and immunity were found between the HGG patients and healthy controls, and also, among the patients with different tumor grades [36].

The relationship between brain activity and the growing tumor in HGG patients has been investigated using MRI by Numan and co-workers (Figure 3c–e) [37]. Postoperative magnetoencephalography was recorded in 45 diffuse glioma patients, and the tumors were segmented on MRI. Brain activity was studied using three measures (absolute broadband power, offset, and slope) and calculated at three spatial levels: the global average, a value averaged across the peritumoral areas, and a value averaged across the homologues of these peritumoral areas in the contralateral hemisphere. Albeit none of the measures of brain activity correlated with changes in the tumor volume, nor did they differ between patients with increasing versus stable tumor volumes; a significantly higher brain activity level was specifically found in the peritumoral areas than that in the contralateral homologous areas, as determined by paired Wilcoxon signed-rank tests. The source of this peritumoral higher brain activity level remains an unresolved issue.

A mismatch repair deficiency (MMRD) can be induced by sporadic or germline mutations and cause a phenotype of predisposition to different forms of neoplasia, including HGG. It has been suggested that a diagnosis of HGG associated with developmental and cerebral cortex abnormalities may lead to the suspicion of an MMRD tumor. A diagnosis of MMRD HGG may have important consequences on the therapy of those who are affected and their families [38]. Thirteen MMRD-associated (nine sporadic and four germline) primary brain tumors have been characterized by Kim and co-workers to determine their clinicopathological and molecular characteristics and biological behavior [39]. The most common MMRD primary brain tumor was GB *IDH* wild-type one. The gene expression profile of this tumor type was different from that of non-MMRD GBs, with a probable effect of resistance to alkylating agents and sensitivity to immunotherapy [39]. The number of nonsense mutations were increased in MMRD tumors compared with those in non-MMRD tumors (Figure 3f).

Increased consumption of glutamine has been observed in some HGGs, which has been correlated with resistance to treatments. Ekici and collaborators discussed this aspect of HGG metabolism, together with the techniques currently available for measuring glutamine levels in the CNS [48].

When the brain tumor is located in the vicinity of eloquent areas, its resection is particularly delicate due to the risk of failing to preserve adequate neurological function. Tuleasca and collaborators discussed the utility of awake surgery coupled with intraoperative MRI to improve the outcome of this particularly challenging type of surgery [49]. The different diagnostic techniques that can be used during surgery to obtain anatomical information and direct the operative technique have been also discussed by Matsumae and collaborators [50]. In particular, the brain stem is the site of the regulation of cardiac and respiratory functions, and given those critical functions, it is difficult to obtain a brain stem biopsy for histopathological diagnostic purposes. Jung and colleagues reported their relevant surgical activity over the past 6 years, suggesting appropriate measures that can reduce the risk associated with performing a stereotaxic biopsy in that important anatomical area [40].

Radiation-induced organizing hematoma (RIOH) is a sporadic form of cavernous hemangioma (CH) that occurs after cerebral radiation. RIOH lesions are distinct histologically from ex novo CH in unirradiated patients. The clinical and histological features of RIOHs were investigated by Kim et al. [41] (Figure 3g–k). The cases of RIOH were divided into two groups based on the RT protocol they had undergone (conventional RT or gamma knife surgery (GKS)). Histological analysis showed a hematoma-like area in the RIOH with a small number of thin-walled vessels, whose walls were infiltrated by fibrin and macrophages (Figure 3g,h). In contrast, ex novo CH was characterized by a thick wall that had not been infiltrated by macrophages (Figure 3i). The onset latency of these lesions was significantly shorter after treatment with GKS compared to that of patients who underwent the treatment with conventional RT. Furthermore, the RIOH in the GKS-treated group had a significantly thicker wall than that in the conventional RT-treated group (Figure 3j–k). RIOH lesion size tended to be larger in the GKS-treated patients group compared to that with conventional RT. Further, RIOH was more likely to occur in subjects who underwent GKS than it was in patients who underwent conventional RT.

## 4. Advances in Prognosis

The storkhead box 1 (STOX1) protein is related to the forkhead family of transcription factors, for which a role in the CNS development and also in the onset of HGG has been suggested [51]. To explore the correlation between STOX1 expression and the prognosis of glioma patients, Jin and co-workers analyzed Kaplan–Meier survival curves together with a log-rank comparison to investigate the differences in the OS of low- and high-level STOX1 glioma patients in each dataset [52] (Table 2). Considering both glioma types and GB alone, patients with high-level STOX1 expression had a significantly higher OS than those with low-level STOX1 expression did, after analysis of both the Chinese Glioma Genome Atlas (CGGA) and The Cancer Genome Atlas (TCGA) datasets (Figure 4a–d). The level of expression of STOX1 might depend on the tumor grade; thus, it possibly represents an additional diagnostic and prognostic factor for HGG [52]. A number of different genetic conditions affect the pharmacokinetics and efficacy of TMZ [53]. In particular, the DNA repair protein, O^6^-methylguanine-DNA methyltransferase (MGMT), causes the resistance of tumor cells to alkylating agents, including TMZ. Although methylation of the *MGMT* promoter is an important diagnostic and prognostic factor for HGG, the technical characteristics of its determination (CpG sites to be analyzed, cut-off values, etc.) are not standardized. Brandner and collaborators published a meta-analysis of 190 published studies on the subject, without being able to identify a consensus. It was not possible to draw conclusions about the quality of the results of methylation analyses obtained using frozen rather than paraffin-fixed or formalin-fixed tissues. Similarly, it was not possible to identify the best CpG sites for carrying out these analyses [53]. To investigate the threshold level of methylation of the *MGMT* promoter influencing the therapeutic outcome, patients with HGG undergoing different RT or chemotherapy protocols were stratified by the number of methylated CpG sites in the *MGMT* promoter by Teske and co-workers [54]. A better therapeutic outcome was confirmed in the presence of methylation; in particular, the presence of more than 18 methylated CpG sites correlated with an increase in both the OS (30 vs. 15 months) and radiographically determined PFS (20 vs. 8 months) (Figure 4e,f).

**Table 2 ijms-24-06375-t002:** Highlights of recent advances in the prognosis of HGG.

Major Finding	Experimental System	LE ^1^	Ref
A grade-dependent reduction on storkhead box 1 (*STOX1*) expression in glioma was revealed. STOX1 may be used as a novel predictive molecular biomarker for glioma grading and patients’ OS.	STOX1 expression in glioma was analyzed using three publicly available datasets, including CGGA, TCGA, and Rembrandt.	VIII	[52]
Extent and patterns of methylated CpG sites are similar in GB and *IDH* wild-type astrocytoma with *ptert* mutations. In both tumor entities, higher numbers of methylated CpG sites appear to be associated with a more favorable outcome.	An institutional database was searched for patients with GB defined by histopathology and *IDH* wild-type astrocytoma with promoter of telomerase reverse transcriptase (*ptert*) mutations. O^6^-methylguanine-DNA methyltransferase (*MGMT*) promotor methylation was analyzed using methylation-specific PCR and Sanger sequencing of CpG sites within the *MGMT* promotor region.	VIII	[54]
Positron emission tomography (PET)-guided gross total resection (GTR) improves the OS in patients with HGG. A multimodal imaging approach including FET-PET for surgical planning in newly diagnosed and recurrent tumors may improve the oncological outcome in HGG patients.	Imaging and survival data from patients with primary and recurrent HGGs who underwent FET-PET before surgical resection were retrospectively analyzed.	VIII	[55]
Similar results for the differentiation of HGG progression from treatment-related changes were obtained using different machine learning models. Further, an additional advantage based on TBR values only was found for tumor-to-background ratio (TBR) static and TTP dynamic radiomics over those of the classical analysis.	Eighty-five patients with histologically confirmed HGG who were investigated by dynamic ^18^F-FDOPA PET in two institutions were retrospectively analyzed.	VIII	[56]
GB patients with deep supratentorial extension (DSE) have a lower OS compared to those without DSE. This survival difference appears to be primarily related to the limited surgical extent of resection, owing to the neurological deficits that may be incurred with involvement of eloquent deep brain structures.	The MRI scans of 419 HGG patients who underwent tumor resection were retrospectively analyzed.	VIII	[57]
Quantitative analysis of conventional MRI sequences can demarcate HGG peritumoral region (PTR) from LGG, which is otherwise indistinguishable by visual estimation.	A total of 74 patients were included in the analysis: 42 patients affected by HGG with preoperative scans of tumors, and 32 patients affected by LGG without high-grade features on imaging.	VIII	[58]
Baseline mean K(ep) may be a useful biomarker for predicting the response and stratifying patient outcomes following bevacizumab treatment in patients with recurrent HGG.	Fifty-three patients with recurrent HGG underwent baseline MRI including diffusion tensor imaging (DTI), dynamic contrast enhancement (DCE), and dynamic susceptibility contrast (DSC) before bevacizumab treatment.	VIII	[59]
This is a precious dataset that can be used to relate the visual appearance of the tumor on the scan with the genetic and histological features and to develop automatic segmentation methods.	The Erasmus Glioma Database (EGD) contains structural MRI scans, genetic and histological features (specifying the WHO 2016 subtype), and whole tumor segmentations of patients with glioma.	VIII	[60]
Even after adjusting for known confounders, married patients with HGG and LGG have a higher possibility of having a better outcome. This study highlights the potential significance that intimate support from spouse may improve glioma patients’ survival.	The Surveillance, Epidemiology, and End Results program was used to identify 81,277 patients diagnosed with the most common primary malignant brain tumors, including glioma, ependymoma, and medulloblastoma.	VIII	[61]
With an organized and dedicated multidisciplinary team, adequate outcomes for pediatric brain tumor patients can be achieved in a middle-income country setting. The presence of local residual disease after surgery and disseminated disease has a strong negative effect on OS.	A series of 173 pediatric patients treated in a Serbian referral oncology institution were analyzed based on their clinical, histological, treatment, and follow-up data.	VIII	[62]
The rs7853346 polymorphism in long non coding (lnc) RNA-*PTENP1* and the rs1799864 polymorphism in chemokine receptor (*CCR)2* could independently affect cognitive impairment after RT, while a more significant combined effect on cognitive impairment was exerted in glioma patients via the signaling pathway of *PTENP1*/miR-19b/*CCR2*.	Two hundred and seventy-nine glioma patients were recruited and grouped according to their genotypes of rs7853346 in *PTENP1* and rs1799864 in *CCR1*.	VIII	[63]
A high incidence of early-onset endocrine disorders was found in examined brain tumor pediatric patients. An endocrine consultation and nutritional evaluation should be mandatory for all patients with a brain tumor, especially when the tumor is suprasellar or after hypothalamus/pituitary irradiation.	This is a noninterventional observational study based on data collection from medical records of 221 brain tumor pediatric patients. The main tumor types were medulloblastoma (37.6%), craniopharyngioma (29.0%), and glioma (20.4%).	VIII	[64]

^1^ Level of evidence according to https://guides.library.stonybrook.edu/evidence-based-medicine/levels_of_evidence (accessed on 14 March 2023).

The response to the treatment of solid tumors is usually assessed according to the RECIST criteria (Response Evaluation Criteria in Solid Tumors) [65]. Those criteria are generally based on the reduction of tumor size in response to treatment. In several cases, however, this dimensional assessment might not be sufficient to adequately evaluate the response to therapies. It has been observed, especially with therapies targeting molecular signaling pathways, that tumor necrosis or the lack of tumor progression may be associated with a good therapeutic response, even in the absence of the size reduction of the neoplasm. The RECIST criteria should, therefore, be integrated with additional biomarkers for assessing the viability and metabolic activity of the tumor that may be related to an early response to therapies and contribute complementary information on the treatment’s efficacy [65].

Whether a gross total resection (GTR) of the biological tumor volume, defined as <1 cm^3^ of residual volume (PET-guided GTR), correlates with better outcomes for the patients was investigated by Ort and co-workers [55]. Patients undergoing PET-guided GTR at initial or recurrent resections showed longer OS, with a median of 19.3 months versus 13.7 months for patients with residual PET (Figure 4g). No significant differences were found between the two patient groups for a number of parameters that can influence the OS, including the age at diagnosis, WHO classification, histopathology, *MGMT* methylation status, *idh* mutation rate, and preoperative tumor volume. Patients who had PET-guided GTR at the initial resection only also showed a survival benefit (17.3 versus 13.7 months, respectively (Figure 4h)) [55].

The relevance of ^18^F-DOPA PET static and dynamic radiomics for the differentiation of HGG progression from treatment-related changes has been evaluated by Ahrari and co-workers [56]. An additional value for such differentiation was provided by the time-to-peak (TTP) dynamic radiomics over the classical analysis based on tumor to back-ground (TBR) values [56].

When the HGGs extend over the tentorium, they can involve the thalamus, the basal ganglia, and the corpus collosum, which are deep structures that are difficult to treat surgically, since any damage to them may cause heavy neurological repercussions [57]. HGG patients, therefore, often have a shorter survival than the others do, due to more limited tumor resection. After Kaplan–Meier analysis, patients with deep supratentorial extension (DSE) showed a significantly reduced survival than those not affected by DSE did (Figure 4i). The worsening of the OS was progressively more pronounced with the increase in the number of deep supratentorial structures (the thalamus, basal ganglia, and corpus callosum) involved (Figure 4j).

The peritumoral region of HGG appears to have T2W hyperintensity and is composed of a microscopic tumor and edema. Infiltrative low-grade gliomas (LGG) comprise tumor cells that seem similar to the HGG peritumoral region on MRI. Whether the HGG peritumoral region with edema can be distinguished from LGG tumor tissues by radiomics has been investigated by Malik and co-workers [58]. It was proposed that the HGG peritumoral regions might be distinguished from LGG through a focused quantitative analysis of MRI images, which is not possible only visually, but requires specific classifiers.

The mean rate transfer coefficient (K(ep)) represents the transit between the extravascular and the intravascular compartments and is known to reflect the vessel permeability and surface area. High vessel permeability is associated with a more malignant phenotype that is resistant to bevacizumab treatment and may be linked to unfavorable treatment outcomes [66]. It was suggested that K(ep) may be a useful biomarker for predicting response and stratifying outcomes following bevacizumab treatment in patients with recurrent HGG [59].

The pre-operative MRI data of hundreds of patients with glioma treated at the Erasmus Medical Center in Rotterdam (NL) have been made available with the Erasmus Glioma Database (EGD) [60]. This precious directory contains MRI investigations and genetic and histological characteristics specifying the WHO subtype. In particular, T1-weighted pre-contrast FLAIR scans are available for all patients, as well as post-contrast images in T1 and T2, obtained using different scanners from four different companies. All the images are stored in a dedicated database. Genetic and histological data consist of the *idh* mutation status, *1p/19q* co-deletion status, and WHO grade. The complete WHO HGG subtype is also available [60]. This database may be useful for relating the morphological characteristics of the tumor with the genetic and molecular profiles, allowing the possible identification of innovative diagnostic and prognostic parameters.

The impact of marital status on the mortality of patients with primary malignant brain tumors has been investigated by Deng and co-workers [61]. After adjusting for known confounders, married patients with HGG have a higher possibility of having a better outcome, indicating the potential significance that intimate support from a spouse can improve HGG patients’ survival.

The characteristics of children with primary brain tumors, the effectiveness of treatment modalities, and the factors related to their outcome have been reported by Stanic and co-workers [62]. The mean survival time of all the children was 94.5 months. The mean probabilities of OS at 2, 5, and 10 years were 68.8, 59.4, and 52.8%, respectively (Figure 4k). As expected, the histopathological type of tumor was the most important predictor of survival. Patients with unknown histopathology (e.g., unresectable brain stem glioma) and HGG had lower survival rates than those with the other histological types did: embryonic tumors, ependymomas, and LGG (Figure 4l). As expected, the presence of a residual disease after exeresis and a disseminated disease resulted in significant prognostic worsening. The authors concluded that the coordinated action of different specialists can obtain good results in the treatment of pediatric HGG, even in a middle-income country such as Serbia.

The effects of long non-coding (lnc) RNA-PTENP1 mRNA and chemokine receptor-2 (CCR2) polymorphisms on cognitive impairment in HGG patients was investigated by Yang and colleagues [63]. It was found that the rs7853346 polymorphism in lncRNA-PTENP1 and the rs1799864 polymorphism in CCR2 may influence cognitive impairment much more if they are present concomitantly. In subsequent studies by the same group of researchers, it was found that rs4702 polymorphism in FURIN may also affect the cognitive ability of HGG patients. In particular, it was hypothesized that carriers of the A allele of the rs4702 polymorphism in FURIN may have a lower risk of cognitive impairment than those carrying the G allele [67].

The effects of HGG and its therapies on the patient’s endocrinology have been discussed by González Briceño and collaborators [64]. Early-onset endocrine disorders are common in HGG patients. Endocrinological and nutritional counseling should be routinely offered to the patient with HGG, especially when the tumor is suprasellar or when the pituitary and/or hypothalamus has been irradiated.

## 5. Concluding Remarks and Future Perspectives

Diagnostic hypotheses of HGG are typically initially assessed by MRI or CT. Under optimal organization conditions, those images allow a neuro-oncological disease management team (DMT) to perform initial classification of the tumor and plan its possible treatment [68,69]. These plans are often complicated by uncertainties relating to the natural history of the tumor (e.g., whether or not it is due to the evolution of lower-grade lesions) and by the presence of outcomes of any previous treatments (e.g., necrosis or pseudoprogression) [70,71]. In the last decade, some molecular characteristics have been identified, making the diagnosis and prognosis of HGG more precise [72,73]. Molecular genetic analysis has become important not only for a correct classification, but also to monitor the progression of the tumor and better manage its therapy. An increasing number of these molecular markers are entering clinical practice, modifying guidelines and suggesting new targeted molecular therapies. In this context, the use of artificial intelligence (AI) techniques in image analysis (MRI, CT, or PET) have contributed improvements to the diagnosis and treatment of HGG, and more generally, to the research on those hard-to-treat tumors. For example, that approach helped to differentiate HGG from other neoplastic lesions, such as lymphomas and metastases, from inflammatory and demyelinating non-neoplastic lesions.

The discipline that uses AI for the calculation, identification, and extraction of image characteristics, as well as the generation of diagnostic and/or prognostic mathematical models, is called radiomics [74]. Radiogenomics allows the combined use of molecular genetics and radiomics data to further improve the accuracy of the diagnostic/prognostic profile [75]. As previously mentioned, the detection of mutations in a key metabolic enzyme (*idh*), histone genes (*h3-3a*), and large-scale chromosomal aberrations (*+7/−10*, *1p/19q*) are examples of specific alterations that importantly integrate the traditional information provided by histopathological analysis [76,77]. Already, the fourth WHO edition of the classification of CNS tumors published in 2016 integrated the traditional histological classification with some parameters of molecular genetics, innovating the way in which oncologists classify CNS tumors and plan their treatment. The fifth edition of the classification of CNS tumors published in 2021 was further enriched with molecular genetic parameters that are useful for a better diagnostic and prognostic classification of HGG in the pediatric field [78,79,80,81]. It is easy to predict that the sixth edition of the classification, which will possibly be available in 2026, will provide the neuro-oncology community with an even more detailed description, in molecular terms, of the different types and subtypes of HGG, and more generally, of brain tumors. It will also probably include the first validated radiomic and radiogenomic analyses. Further, HGGs release traces of themselves in body fluids, such as the bloodstream or cerebrospinal fluid (CSF), and scientific interest is growing in this molecular information that is easily obtainable from liquid biopsies that can possibly surrogate the tumor tissue [82]. Examples of those biomarkers include nucleic acids such as microRNAs, proteins, and tumor-derived extracellular vesicles. In recent years, cancer cells circulating in the blood of patients with HGG have also been identified. All those new diagnostic and prognostic tools, once validated, may enter new editions of CNS tumor classification.

A systematic review of the efficacy of systemic therapies for recurrent HGG in adults was recently published [83]. A total of 48 randomized controlled trials were evaluated. Since the therapeutic outcomes reported in the various studies were heterogeneous (the OS, PFS, and OS at 6 months, PFS at 6 months, and objective response rate), it was not possible to perform a meta-analysis. The median OS across all studies ranged from 3 to 17.6 months, and no treatment was significantly superior to the others. Hence, there is currently no systemic treatment capable of significantly improving the OS of HGG patients compared to that which the standard “Stupp” radio/chemotherapeutic protocol can induce [83], and 18 years after the introduction of temozolomide, the development of new therapies is still dramatically slow [84]. This deadlock is, at least in part, linked to the high (multiform) heterogeneity of HGG at all levels: morphological, histological, and molecular levels. Furthermore, the microenvironment in which the tumor grows is also an important player in the progression of the disease, and it is another factor of heterogeneity and resistance. The active communication between the glioma cells and local or nearby healthy cells and the immune environment promotes the dominance of tumor populations with the characteristics of increasing malignancy and resistance to therapies, which are often associated with the expression of stem markers. Strengthening research on the stem component of HGG tumors, which currently appears to be the most malignant and therapy-resistant target, could be a fruitful choice [85].

As aforementioned, radiomics and radiogenomics may help with better defining the effectiveness of the different treatments (radio- and chemotherapeutic and immunological ones), since these computational methods have increased the comparability of the data, and analysis has become more independent from the subjectivity of the clinician [74]. Further, the characterization of tumor infiltration based on preoperative multiparametric MRI (MP-MRI) might predict the loci of the most probable future recurrence, permitting clinicians to anticipate their focused treatment [86,87]. The hope is that developments in the genetic classification of HGGs and the use of predictive models based on radiomics may provide important advances in tumor characterization and in the personalization of therapies. However, in the case of HGG, no radiomics technique has yet entered clinical practice, given their still experimental character and the heterogeneity of the designs, software, and equipment used. AI techniques have, until now, been applied using different parameters and methodologies, and their comparison and validation, in general terms, are not yet possible. Further, most of the published studies are based on small datasets, and they often lack external validation data [88,89]. Carefully controlled multicenter clinical trials involving heterogeneous study populations are required so that radiomics and radiogenomics can reach the degree of reliability that is necessary to be included in clinical practice [68].

In conclusion, albeit the evidence is still inevitably limited for most major findings reviewed here (VIII), since they result from single institutional studies (Table 1 and Table 2), the advancement in the diagnosis and prognosis of HGG has been gradual, but steady, as witnessed by the quantity and quality of research published in just a short time and briefly discussed here, as well as the regular updating of WHO classifications of human brain tumors [9,90]. Unfortunately, the improvements in our diagnostic and prognostic capacity for HGG have not had a significant impact on patient treatment and survival so far; no significant progress in increasing the median OS of HGG patients has been recorded in the past 18 years (the last effective study was published in 2005 [4,6]). Hence, one most important future direction of the neuro-oncology community should be to investigate the causes of those two-speed achievements of research and fill the gap.

## Figures and Tables

**Figure 1 ijms-24-06375-f001:**
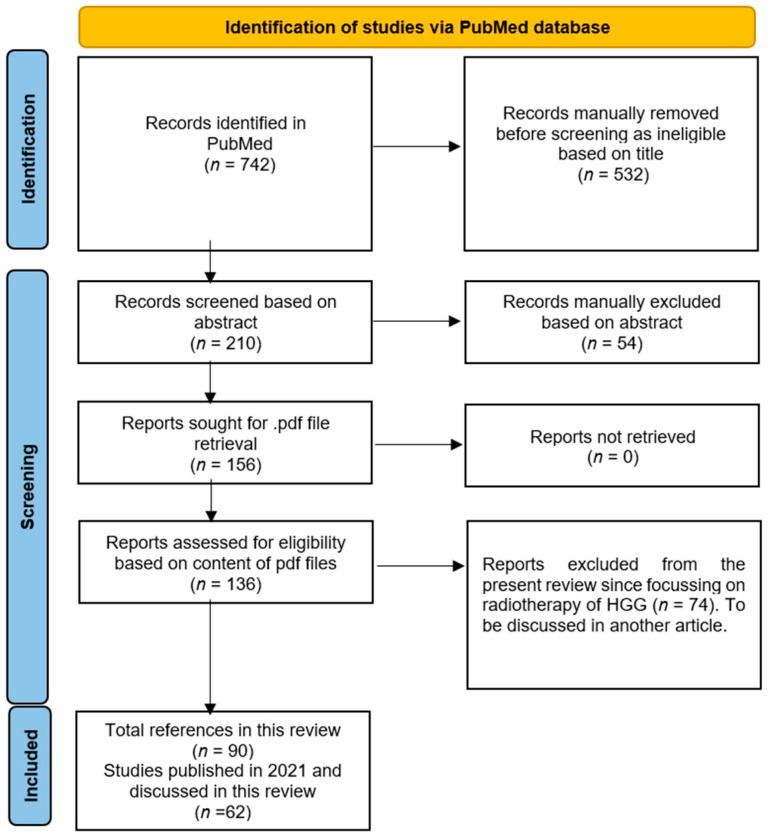
Flow diagram for reported studies (modified from [7] with permission).

**Figure 2 ijms-24-06375-f002:**
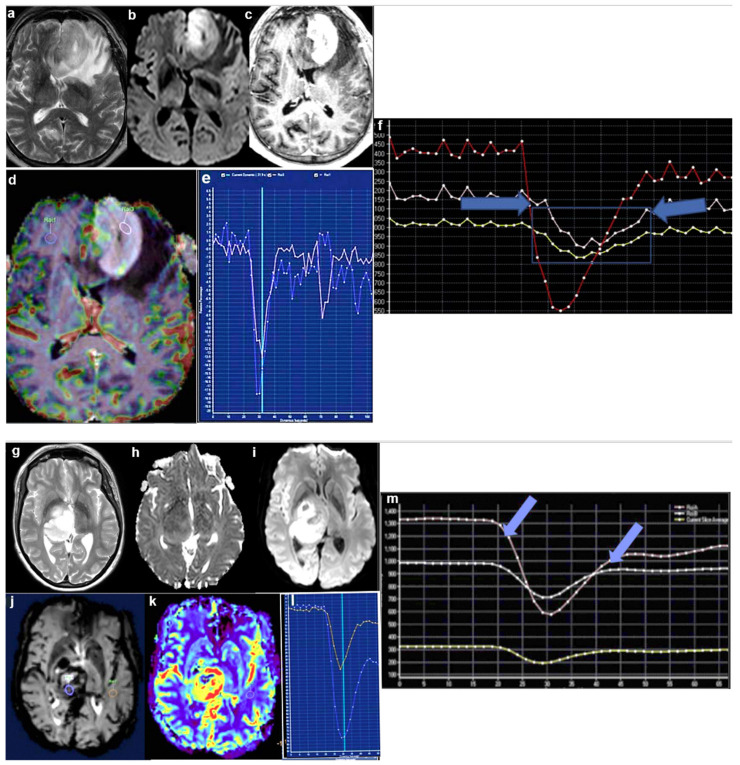
Advances in imaging diagnosis of HGG. PCNSL. (**a**). Axial T-2. (**b**). DWI. (**c**). Postcontrast studies demonstrate an intensely enhancing mass in the left frontal lobe with a mild T-2 dark signal and T-2 shortening, demonstrating DWI restriction and marked contrast enhancement. (**d**,**e**). Co-registered perfusion with postcontrast T1 volumetric and perfusion curves demonstrate low blood volume relative to white matter in spite of significant enhancement in the postcontrast studies and a significantly higher percentage signal recovery than that of normal white matter (pink region of interest (ROI) representing tumor and blue ROI normal white matter). Percentage signal recovery curve showing that is characteristically higher than base line recovery is. (**f**). Magnified version of perfusion graph, highlighting the difference in percentage signal recovery values between PCNSL and normal white matter (pink = tumor; yellow = white matter; red = mean of entire slice). GB. (**g**). Axial T2. (**h**). DWI (**i**). Apparent diffusion coefficient (ADC): there is a mass of T2 heterogenous intensity within the right basal ganglia, with DWI restriction and low ADC, closely resembling lymphoma. (**j**–**l**). Perfusion study demonstrating high blood volume and low PSR. In contrast to PCNSL, GB demonstrates significantly lower percentage signal recovery than normal white matter does. (**m**). Magnified version of PSR (blue arrows), showing the difference of signal recovery values between the tumor and white matter (pink = tumor; white = white matter; yellow = mean of tissue slice) (reproduced from [15] with permission). Typical localizations of pDMG in the pons (**n**,**o**) and thalamus (**p**,**q**) in axial MR images. Different *H3K27* genetic subgroups may show very similar MRI phenotypes with hyperintense, heterogeneous T2 signal (**n**–**q**). However, T2 signal may also appear to be very different even within the same genetic group (**r**,**s**) (reproduced from [16] with permission). MRI and histopathological staining of brain tumor tissues. (**t**). Sagittal T1-weighted contrast-enhanced MRI imaging of initial and recurrent tumors. (**u**,**v**). Representative images of u. primary GB and (**v**). secondary gliosarcoma (SGS) tissues stained using hematoxylin and eosin (HE), and immunolabeled using antibodies to glial fibrillary acidic protein (GFAP) and reticulin (reproduced from [17] with permission).

**Figure 3 ijms-24-06375-f003:**
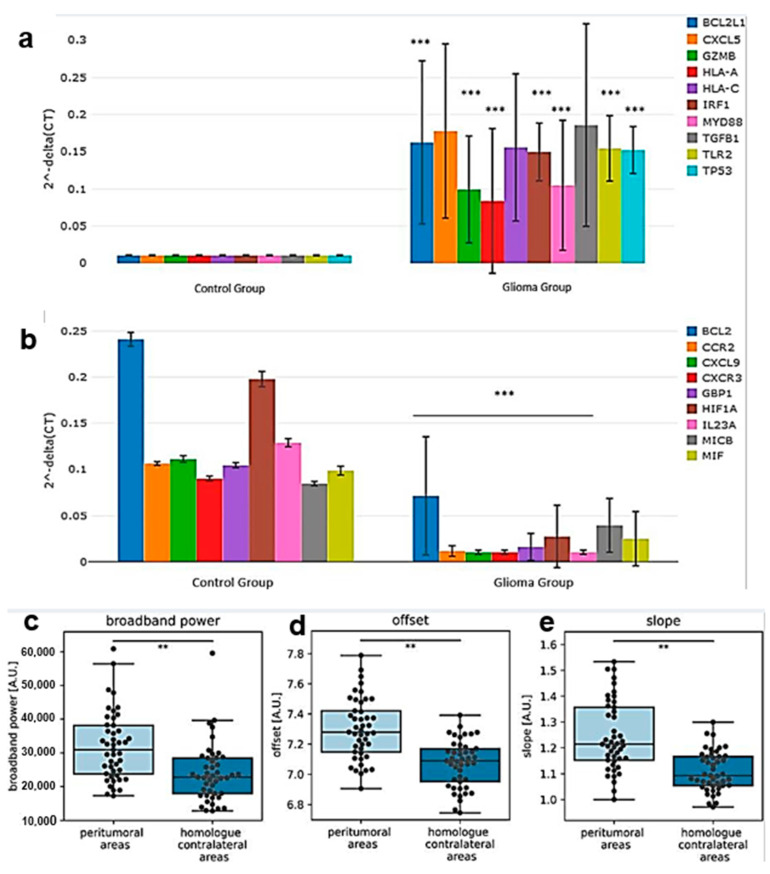
Advances in non-imaging diagnosis of HGG. (**a**). Column chart demonstrating significantly over-expressed genes in the plasma samples of glioma patients relative to those of healthy control. (**b**). Column chart demonstrating significantly under-expressed genes in the plasma samples of glioma patients relative to those of healthy control (reproduced from [36] with permission). (**c**–**e**). Brain activity of peritumoral and homologue contralateral areas. Peritumoral brain activity level was significantly higher compared to the level in the homologue contralateral areas. *p* < 0.001 after correction for three comparisons (**c:** broadband power; **d:** offset; **e:** slope). A.U. = arbitrary units (reproduced from [37] with permission). (**f**). The box plot of the number of nonsense mutation of HGG with/without mismatch repair deficiency (MMRD). MMRD HGG had a higher number of nonsense mutation than non-MMRD HGG did. The average numbers of mutations in astrocytoma *idh* mutant with MMRD and without MMRD were 23.0 and 5.8, respectively. The average numbers of mutations in *IDH* wild-type GB with MMRD and without MMRD were 23.6 and 4.7, respectively (reproduced from [39] with permission). (**g**–**i**). The histology of radiation-induced organizing hematoma (RIOH) and ex novo cavernous hemangioma (CH). Microscopically, RIOH shows a hematoma-like area composed of hyalinized vessels with fibrin and infiltrating foamy macrophages (**g**,**h**) and ex novo CH consists of clusters of well-formed vascular lumens (**i**). (**j**,**k**). Gamma knife surgery-induced RIOH shows relatively thicker tumor walls (**j**) compared with those of conventional RT-induced RIOH (**k**) (Masson’s trichrome) (reproduced from [41] with permission). Asterisks indicate the level of statistical significance.

**Figure 4 ijms-24-06375-f004:**
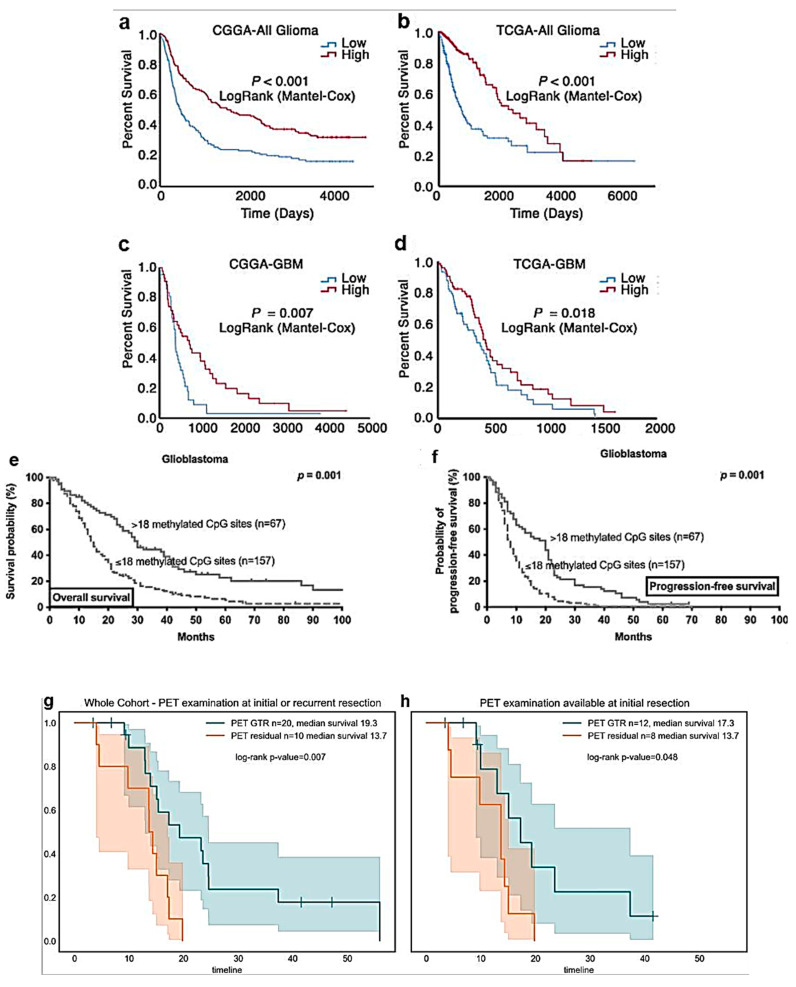
Advances in prognosis of HGG. Kaplan–Meier analysis of OS based on STOX1 expression levels in glioma patients. (**a**–**d**). Glioma patients with high-level STOX1 expression had longer OS than those with low-level STOX1 expression did among the whole glioma cohort (**a**,**b**) or GB cohort alone (**c**,**d**) in both CGGA and TCGA datasets (reproduced from [52] with permission). (**e**,**f**). *MGMT* is a marker for survival and disease progression in GB. (**e**,**f**). The extent, pattern, and prognostic value of *MGMT* promotor methylation. (**e**). Kaplan–Meier estimates of OS in GB treated with any form of radio-/chemotherapy. Curves are displayed for patients with >18 methylated CpG sites (straight lines) and ≤18 methylated CpG sites (dotted lines). (**f**). Kaplan–Meier estimates of radiographic progression-free survival in GB treated with any form of radio-/chemotherapy. Curves are displayed of patients with >18 methylated CpG sites (straight lines) and ≤18 methylated CpG sites (dotted lines) (reproduced from [54] with permission). (**g**,**h**). In HGG patients, ^18^F-FET-PET-guided gross total resection (GTR) improves OS. (**g**). PET GTR results in longer OS (19.3 months) compared to that of patients with PET residual (13.7 months). (**h**). Patients with FET-PET examination before initial resection showing a significant effect of PET GTR (reproduced from [55] with permission). (**i**,**j**). HGG with deep supratentorial extension (DSE) is associated with a worse OS. (**i**). Kaplan–Meier curves reveal that GBs with DSE are associated with significantly worse OS compared to those without DSE. (**j**). Involvement of a greater number of deep-seated brain structures is associated with progressively worsening OS by pooled log-rank analysis (reproduced from [57] with permission). (**k**,**l**). Clinical profile, treatment, and outcome of pediatric brain tumors in Serbia over a 10-year period. (**k**). Cumulative survival of children with brain tumors. (**l**). Cumulative survival of children with brain tumors according to tumor histopathological type. ET—embryonal tumors; HGG—high-grade glioma; LGG—low-grade glioma; EP—ependymoma; UH—unknown histopathological type (reproduced from [62] with permission).

## Data Availability

Data sharing is not applicable to this article.

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
