# Peer review of "Recapitulating the Key Advances in the Diagnosis and Prognosis of High-Grade Gliomas: Second Half of 2021 Update"

_ijms, 2023, doi:10.3390/ijms24076375_

Round 1

Reviewer 1 Report

In "Recapitulating the Key Advances in the Diagnosis and Prognosis of High-Grade Gliomas: second half of 2021 update" the authors comprehensively summarise the later 2021's semester literature on HGG's diagnosis, prognosis, and overall survival. 
I just had a few remarks or notes, as per the attached file. Otherwise, this manuscript is ready and suited to be published in IJMS.

Author Response

                                                                                 Genova, March 2, 2023

Dear Editors,

please find enclosed the revised (R1) manuscript ijms-2183983 entitled “Recapitulating the Key Advances in the Diagnosis and Prognosis of High-Grade Gliomas: second half of 2021 update” by Guido Frosina for submission to the International Journal of Molecular Sciences - Special Issue “Glioblastoma: Recapitulating the Key Breakthroughs and Future Perspective 2.0” as a Review article.

The useful comments and suggestions by Reviewer 1 were dealt with as following:

Reviewer 1

 In "Recapitulating the Key Advances in the Diagnosis and Prognosis of High-Grade Gliomas: second half of 2021 update" the authors comprehensively summarize the later 2021's semester literature on HGG's diagnosis, prognosis, and overall survival.

I just had a few remarks or notes, as per the attached file. Otherwise, this manuscript is ready and suited to be published in IJMS.

The insightful remarks or notes, in the file attached by Reviewer 1 were all dealt with and highlighted in red. Briefly:

The section “2. Methods” was partially re-written, to make clearer the workflow for the identification of relevant studies. In particular, the .pdf files of the 156 articles resulting from the initial selection steps based on titles and abstracts, were all retrieved. Based on the contents of the .pdf full-articles, 20 of them were discarded as out of scope while the remaining 136 were further analyzed and manually assigned to the "Radiology of HGG" group (n = 62, discussed here) or " Radiotherapy of HGG" group (n = 74, to be discussed in another article). In addition to the 62 articles published in 2021, for the purpose of clarity and completeness, this review includes further 28 relevant articles published outside the year 2021 for a total of 90 references. The relevant flowchart shown in Figure 1 was modified accordingly.

Figures 1-4. Quality of all Figures was improved and the font size increased so that they can be more easily read. The ROIs in Figure 2d are identified by small circles. Some weak color differences (e.g. between pink and white) in Figure 2 plots relate to the originally published pictures and could not be further enhanced.

Table 1. The acronyms requiring definition were written in full.

Lines 306-307. The validating statistical analysis employed was indicated (paired Wilcoxon signed-rank test).

Two novel references ([43], [51]) were inserted.

Sincerely

Guido Frosina PhD

Mutagenesis & Cancer Prevention Unit

IRCCS Ospedale Policlinico San Martino

Istituto di Ricovero e Cura a Carattere Scientifico

Largo Rosanna Benzi 10

16132 Genova, Italy.

Tel.: +39.010.5558543

FAX: +39.010. 5558237

E-mail: guido.frosina@hsanmartino.it

Reviewer 2 Report

This review, summarizes publications related to gliomas within the timefram from July 2021 to December 2021. If indicates potential advancements mainly in the fields of pathology, imaging and circulating disease-associated markers. The review is written very well.

Major

Quality of Figure 3 should be improved. The font size should be increased as its hard to read in its current version

The evidence level of observations mentioned in the tables should be provided so that the readership can judge the level of evidence and I expect the evidence level pretty low for most statements, as they result from single institutional studies.

Author Response

                                                                         Genova, March 2, 2023

Dear Editors,

please find enclosed the revised (R1) manuscript ijms-2183983 entitled “Recapitulating the Key Advances in the Diagnosis and Prognosis of High-Grade Gliomas: second half of 2021 update” by Guido Frosina for submission to the International Journal of Molecular Sciences - Special Issue “Glioblastoma: Recapitulating the Key Breakthroughs and Future Perspective 2.0” as a Review article.

The useful comments and suggestions by Reviewer 2 were dealt with as following:

 Reviewer 2

This review, summarizes publications related to gliomas within the timeframe from July 2021 to December 2021. If indicates potential advancements mainly in the fields of pathology, imaging and circulating disease-associated markers. The review is written very well.

 Major

 Quality of Figure 3 should be improved. The font size should be increased as it’s hard to read in its current version.

Figures 1-4. Quality of all Figures was improved and the font size increased so that they can be more easily read. The ROIs in Figure 2d are identified by small circles. Some weak color differences (e.g. between pink and white) in Figure 2 plots relate to the originally published pictures and could not be further enhanced.

The evidence level of observations mentioned in the tables should be provided so that the readership can judge the level of evidence and I expect the evidence level pretty low for most statements, as they result from single institutional studies.

 Tables 1-2. The evidence level of observations mentioned in the tables was provided so that the readership can judge the level of evidence. The Level of Evidence scale by The Stony Brook University NY, https://guides.library.stonybrook.edu/evidence-based-medicine/levels_of_evidence  was used. Indeed, the evidence level was pretty low for most statements (VIII), since they result from single institutional studies.

Lines 581-587. Since the evidence level of observations mentioned in the Tables was provided, the concluding remarks were modified as following: “In conclusion, albeit the evidence level still is inevitably low for most major findings reviewed here (VIII) since they result from single institutional studies (Tables 1-2),  the advancement in the diagnosis and prognosis of HGG is gradual but steady as witnessed by the quantity and quality of research published in just a semester and briefly discussed here, as well as the regular updating of WHO classifications of human brain tumors [90] [9]. Unfortunately, the improvements in our diagnostic and prognostic capacity for HGG have not had a significant impact on patient treatment and survival so far: no significant progress in increasing median OS of HGG patients was recorded over the past 18 years (last effective study published in 2005 [4] [6]). Hence, one most important future direction of the neuro-oncology community should be to investigate the causes of those two-speed achievements of research and fill the gap.”

Sincerely

Guido Frosina PhD

Mutagenesis & Cancer Prevention Unit

IRCCS Ospedale Policlinico San Martino

Istituto di Ricovero e Cura a Carattere Scientifico

Largo Rosanna Benzi 10

16132 Genova, Italy.

Tel.: +39.010.5558543

FAX: +39.010. 5558237

E-mail: guido.frosina@hsanmartino.it

Reviewer 3 Report

In this review, the author aims at giving an overview about advances in new imaging modalities and techniques to preoperatively assess and characterize more precisely the possible tumor entities by perfusion, susceptibility changes among others. In this sense, the author systematically points out several findings from studies on MRI in eg. HGGs vs lymphomas. Similarly, results and differences for different PET tracers are discussed.

Additionally, important factors for prognosis in HGGs are discussed.

Overall, the summary these findings and advances is of interest, however, thorough corrections to improve readability and shorten some passages could increase the quality of the paper.

Round 2

Reviewer 2 Report

The authors addressed all points that I raised sufficiently.